# Meal Patterns and Changes in Cardiometabolic Risk Factors in Children: A Longitudinal Analysis

**DOI:** 10.3390/nu12030799

**Published:** 2020-03-18

**Authors:** Xianwen Shang, Yanping Li, Haiquan Xu, Qian Zhang, Ailing Liu, Songming Du, Guansheng Ma

**Affiliations:** 1School of Behavioural and Health Sciences, Australian Catholic University, East Melbourne, VIC 3002, Australia; xianwen.shang@unimelb.edu.au; 2Department of Medicine (Royal Melbourne Hospital), University of Melbourne, Parkville, VIC 3010, Australia; 3Department of Nutrition, Harvard T. H. Chan School of Public Health, 655 Huntington Ave, Boston, MA 02115, USA; yanping@hsph.harvard.edu; 4Institute of food and nutrition development, Ministry of Agriculture and Rural Affairs, Beijing 100081, China; xuhaiquan@caas.cn; 5National Institute for Nutrition and Health, Chinese Center for Disease Control and Prevention, Beijing 100050, China; zhangqian7208@163.com (Q.Z.); liuailing72@126.com (A.L.); 6Chinese Nutrition Society, Beijing 100022, China; dusm9709@126.com; 7Department of Nutrition and Food Hygiene, School of Public Health, Peking University, Beijing 100191, China

**Keywords:** meal pattern, energy, carbohydrate, protein, fat, cardiometabolic risk factors

## Abstract

We examined whether energy and macronutrient intake from different meals was associated with changes in cardiometabolic risk (CMR) factors in children. CMR score (CMRS) was computed by summing Z-scores of waist circumference, the average of systolic and diastolic blood pressure, fasting glucose, high-density lipoprotein cholesterol (multiplying by −1), and triglycerides. We included 5517 children aged 6–13 years from six major cities in China. Five meal patterns were identified according to energy intake: balanced, breakfast dominant, lunch dominant, dinner dominant, and snack dominant patterns. These patterns were not significantly associated with changes in CMR factors. Carbohydrate intake (% energy) at lunch was positively associated with the change in CMRS (beta coefficient (95% CI): (0.777 (0.509, 1.046) in quintile 5 versus quintile 1). A positive association between carbohydrate intake at dinner and change in CMRS was observed. High protein intake at both lunch and dinner was associated with a favorable change in CMRS. Moderate fat intake at lunch was associated with a lower increase in CMRS. Meal patterns driven by energy were not significantly associated with CMR factors; however, a low carbohydrate-high protein-moderate fat lunch and low carbohydrate-high protein dinner were associated with favorable changes in CMRS in children.

## 1. Introduction

There is a high prevalence of cardiometabolic risk (CMR) factors in both children and adults globally [1,2,3], which imposes tremendous burdens on health care and economic systems. It is imperative to prevent these CMR factors in children since childhood CMR factors are highly likely to persist into adulthood and are associated with numerous implications, including hypertension, diabetes, and cardiovascular disease in the future [4,5,6,7,8]. Dietary quality is of paramount importance for the reduction and prevention of CMR factors [9,10,11].

Evidence from recent studies in adults has shown that meal patterns play an important role in the development of CMR factors [12,13]. Eating three meals (breakfast, lunch, and dinner) plus snacks per day is a norm and may be an optimal choice for health [14]. Previous studies have shown that skipping meals, especially breakfast, was associated with a higher prevalence of CMR factors [15,16,17], whereas a higher meal frequency was associated with a lower risk of obesity, high cholesterol, and diabetes in adults [18,19,20]. More recent research in adults has shown that three to four meals per day were associated with lower CMR, compared with one to two, or six or more meals per day [13]. Some studies have demonstrated that a late-night meal is associated with poorer diet quality and adiposity among adults [21,22]. The meal composition of macronutrients may also play an important role in metabolic health, given that diets high in protein and low in carbohydrates are associated with lower CMR in adults [23,24]. However, data on whether meal timing and composition are predictive of changes in CMR factors in children are limited.

The present study examined the association of different meal patterns, driven by energy intake, with changes in CMR factors in a large sample of Chinese children. We also examined whether energy, carbohydrate, protein, and fat consumed at breakfast, lunch, dinner, and as a snack was associated with the change in CMR.

## 2. Materials and Methods 

### 2.1. Participant Selection

The nutrition-based comprehensive intervention study on childhood obesity in China is a multicenter, randomized cluster controlled trial, and the study has been detailed elsewhere [25]. Briefly, the study was conducted in six capital or province capital cities, including Beijing, Shanghai, Chongqing, Jinan, Harbin, and Guangzhou. A total of 9901 children from 390 classes within 38 schools were screened for eligibility. Among the 9867 children who were assessed at baseline (May 2009), 8572 were reassessed at follow-up (May 2010). 5517 children who had dietary intake and CMR markers assessed were included in the final analysis (Appendix A).

The study protocol was approved by the Ethical Review Committee of the National Institute for Nutrition and Food Safety, Chinese Centre for Disease Control and Prevention. Oral assent was collected from children, while written informed consent was obtained from the next of kin, carers, or guardians of all participants.

### 2.2. Dietary Assessment

Dietary intake was assessed using 24 h diet recalls for three consecutive days (two weekdays and one weekend day) in children in grades 2–5. During the interview, samples of local household dishes and utensils (different sizes of bowls, plates, and spoons) were displayed to the children. They were then shown pictures of common foods eaten in these dishes or utensils to indicate portion size consumed. Diet intake was recalled immediately after each meal to make sure of the accuracy of the assessment. The trained interviewer and the tutor would help children recall food intake at school, while parents would help recall foods consumed at home. 

Nutrients and energy intake were calculated based on the China food composition [26]. The average amount of energy and macronutrient intake at breakfast, lunch, dinner, and snack per day was computed. Energy adjusted nutrient consumption per day was computed as 100 × weight (gm)/total energy intake (Kcal).

We defined a balanced meal pattern as ≤12.0% (smallest quartile) difference between the largest and smallest meals of total energy intake. Among the remaining participants, energy intake from snacks greater than 28.5% of total energy intake (largest decile) was defined as a snack dominant meal pattern. We then defined the largest meal, being breakfast, as a breakfast dominant meal pattern, and so on for lunch dominant and dinner dominant meal patterns (Figure 1).

### 2.3. Confounders

Physical activity was assessed using a validated physical activity questionnaire, from which metabolic equivalent (MET) was calculated [27]. Birthweight, household income, parental education, and parental height and weight were reported by parents using a self-administered questionnaire. 

### 2.4. Physical Examinations and Blood Tests

Measurements of physical examinations and blood tests (10–14 h fasting beforehand) were performed at both baseline and follow-up, following standardized procedures by trained staff.

Height was measured to the nearest 0.1 cm using a freestanding audiometer, and weight was measured to the nearest 0.1 kg using a balance-beam scale. Body mass index (BMI) was computed as weight in kilograms divided by the square of height in meters. Waist circumference (WC) was measured midway between the lowest rib and the superior border of the iliac crest on expiration to the nearest 0.1 cm, and the average of the two measurements was used.

Blood pressure was measured in the seated position using a mercury sphygmomanometer (XJ300/40-1, Made in Shanghai) by trained nurses, with at least a 10 min rest before the measurement. The first and fifth Korotkoff sounds were used to represent the systolic and diastolic blood pressure (SBP and DBP). Three measurements were taken to the nearest 2 mmHg, and the average of the last two measurements was used.

A single frequency (50 Hz) hand to foot bioelectrical impendence device (ImpDF50, Impedimed Pty Ltd., Qld, Australia), with subjects in a calm state, was used to assess body composition. Body fat mass was computed using the prediction formula developed by Deurenberg et al. [28], and percent body fat (PBF) was calculated as fat mass divided by body weight.

Fasting glucose was measured using the glucose–oxidize method (Daiichi Pharmaceutical Co., Ltd., Tokyo, Japan) within four hours after the fasting blood sample was obtained. Fasting insulin was measured using the immunoenzymatic method (analyzer AXSYM, Abbott Co., Ltd., Japan). The homeostatic model assessment of insulin resistance (HOMA-IR) was computed as fasting insulin (µU/L) × fasting glucose (mg/dL)/405. 

Conventional enzymatic assays were used to measure levels of serum triglycerides, total cholesterol (TC), high-density lipoprotein cholesterol (HDL-C) and low-density lipoprotein cholesterol (LDL-C) with a 7080 automatic analyzer (Daiichi Pharmaceutical Co., Ltd., Tokyo, Japan).

### 2.5. Statistical Analysis

BMI, WC, PBF, SBP, DBP, TC, HDL-C, LDL-C, TG, insulin, and HOMA-IR at baseline and follow-up were standardized, i.e., Z scores were calculated as Z = (value − mean)/SD, using sex and age-specific means, and SDs. CMR score (CMRS) was calculated by summing Z scores of WC, the average of SBP and DBP, fasting glucose, HDL-C (multiplying by −1), and TG [29].

ANOVA for continuous variables and chi-square tests for categorical variables were performed to compare the difference of baseline characteristics across meal patterns. ANOVA for continuous variables, and chi-square tests for categorical variables, were also conducted to test the difference of baseline characteristics between individuals included in the analysis and excluded from the analysis (due to missing or abnormal diet data).

Since the interaction between sex/intervention and energy and macronutrient intake at different meals for the change in CMRS was not significant (Appendix A), we analyzed the whole population. The general linear regression model (GLM) was used to test the difference in changes in CMR factors between participants with different meal patterns. We tested the following models: (1) age, sex, corresponding CMR factor at baseline as a fixed effect, and clustering effect of classes in schools as a random effect; (2) model 1 plus intervention group, puberty, grade, BMI, physical activity, and energy intake at baseline; (3) model 2 plus birth weight, breastfeeding, household income, or parental BMI and education. Missing values for categorical confounders were assigned as a single category, while means were given to missing values for continuous confounders.

We then analyzed the association between quintiles of energy and macronutrient intake at breakfast, lunch, dinner, and snack; and the change in CMRS using GLM. To control the heterogeneity of foods within the groups of macronutrient intake, major food groups including grains, fried foods, vegetables, fruit, nuts, pork, red meat rather than pork, poultry, eggs, milk, and sugar-sweetened beverages were adjusted for in the multivariable analysis. We used the Benjamin–Hochberg procedure to control the false discovery rate at level 5% for multiple comparisons [30]. Bonferroni P value adjustments were performed for all pairwise comparisons. The association of change in macronutrient intake at each meal, with the change in CMRS, was also analyzed.

Sensitivity analysis was performed for the association between meal patterns and changes in CMR factors in children in the control group. Data analyses were conducted using SAS 9.4 for Windows (SAS Institute Inc.), and all P values were two-sided.

## 3. Results

### 3.1. Baseline Characteristics of Participants

A sample of 5517 children (50.7% boys) aged 6–13 years (mean ± SD; 9.54 ± 1.18) were included in the final analysis. Children excluded from the analysis because of missing or abnormal diet data were more likely to be boys and have higher BMI, WC, fasting glucose, insulin, and lower SBP and DBP compared with those included in the analysis. No significant difference in CMRS was observed between the two groups (Appendix A). 

Dinner dominant meal pattern was associated with higher BMI, SBP, LDL-C, and TG compared with a balanced meal pattern (all *p* values <0.05). No significant difference in energy intake, physical activity, and socioeconomic status was observed between different meal patterns (Table 1 and Appendix A).

### 3.2. Energy and Macronutrients from Meals

Energy consumed from breakfast, lunch, dinner, and snacks was 31.69%, 30.49%, 27.48%, and 10.34%, respectively. Protein intake (% energy of the corresponding meal) from breakfast, lunch, and dinner was 15.30%, 18.69%, and 20.18%, respectively, whereas the number for carbohydrates was 59.03%, 56.39%, and 53.87%, respectively. The energy intake for a balanced meal pattern ranged from 28.49% at dinner to 29.18% at lunch. Children with a breakfast dominant meal pattern consumed 47.01% of their total energy from breakfast, those with a lunch dominant meal pattern consumed 45.58% of their total energy from lunch, those with a dinner dominant meal pattern consumed 44.11% of their total energy from dinner, and those with a snack dominant meal pattern consumed 38.42% of their total energy from snacks (Figure 2). 

### 3.3. Meal Patterns and Changes in CMR Factors

The snack dominant meal pattern was associated with a higher increase in HDL-C compared with the balanced meal pattern. Meal patterns driven by energy intake were not significantly associated with changes in other CMR factors examined before or after adjustment for covariates (Table 2).

### 3.4. Energy Intake from Different Meals and Changes in CMRS

Energy intake from lunch was positively associated with the change in CMRS (β (95% CI): 0.248 (0.044, 0.452) in quintile 5) before but not after adjustment for confounders (0.173 (–0.021, 0.367)). Energy intake from breakfast, dinner, or snacks was not associated with the change in CMRS (Table 3).

### 3.5. Macronutrients Intake at Different Meals and Changes in CMRS

Carbohydrate intake at breakfast was inversely associated with the change in CMRS (Appendix A). This association was attenuated to be nonsignificant after adjustment for the intake of major foods. Children in quintiles 4 (β (95% CI): 0.603 (0.368, 0.837)) and 5 (0.777 (0.509, 1.046)) of carbohydrate intake at lunch had a higher increase in CMRS than those in quintile 1 in the multivariable-analysis. A positive association between carbohydrate intake at dinner and change in CMRS was observed. Carbohydrate intake from snacks was not significantly associated with the change in CMRS.

The multivariable-adjusted β (95% CI) for quintile 5 of protein intake at lunch with the change in CMRS was –0.463 (–0.710, –0.217) (*p*-trend <0.0001). The corresponding number for protein intake at dinner was –0.360 (–0.602, –0.117) (*P*-trend <0.0001). No significant association between protein intake at breakfast or from snack and change in CMRS was observed.

The positive association between fat intake at breakfast and change in CMRS was significant before but not after adjustment for intake of major food groups. Moderate fat intake at lunch (quintile 4) was associated with a lower increase in CMRS. Fat intake at dinner or from snacks was not associated with the change in CMRS (Table 4).

### 3.6. Change in Macronutrients Intake at Different Meals and Changes in CMRS

Increased carbohydrate intake at lunch was associated with a higher increase in CMRS. Increased protein intake at lunch and dinner was each associated with a higher decrease in CMRS. Increased fat intake at breakfast was associated with a lower decrease in CMRS before, but not after, adjustment for confounders (Appendix A). 

### 3.7. Sensitivity Analysis

Similar results for the association of meal patterns with changes in most CMR factors were found in children in the control group (Appendix A).

## 4. Discussion

In this longitudinal analysis with a large sample size of Chinese children, we found meal patterns driven by energy were not associated with changes in CMR factors. Energy intake at lunch was positively associated with the change in CMRS. A low carbohydrate-high protein-moderate fat lunch was inversely associated with the change in CMRS, while a low carbohydrate-high protein dinner was associated with a favorable change in CMRS. 

In China, balanced meal patterns with breakfast, lunch, and dinner contributing 30%, 40%, and 30% of total energy, respectively, were recommended for health. Our study showed that a balanced meal pattern was associated with lower BMI, SBP, LDL-C, and TG compared with a dinner dominant meal pattern in the cross-sectional analysis; this is consistent with previous studies showing that a large dinner was associated with high CMR [21,31]. However, this association was not confirmed in our longitudinal analysis. We found that higher energy intake in different meals was not associated with a higher increase in CMRS. The snack dominant meal pattern was associated with a higher increase in HDL-C, which may be partly attributed to the fact that the majority of snacks were fruits and dairy in our study (data not shown).

Macronutrients such as protein, fat, and carbohydrate contribute to almost all the energy intake, given most children in China do not drink alcohol at all, such that metabolism of macronutrients plays an essential role in cardiometabolic health. Our study supports some previous studies showing that meal composition was associated with a change in CMRS [23,24,32]. We found that high carbohydrate and low fat intake at breakfast were associated with a favorable change in CMRS. This is consistent with a prospective cohort study in adults showing that increasing carbohydrate intake, while simultaneously reducing fat intake at breakfast, was associated with a lower likelihood of metabolic syndrome [33]. Our further analysis demonstrated that this association was not significant after adjustment for the intake of major food groups; this suggests the importance of food intake that consists of macronutrients for metabolic health in children. 

A low-carbohydrate, high-protein, and moderate-fat lunch was associated with a lower increase in CMRS in our study. Evidence has highlighted the importance of protein intake for children’s growth and development, and sufficient protein intake is recommended for each meal [32]. Meanwhile, high protein intake may help enhance satiety and reduce total energy intake [34,35], which may be associated with lower metabolic risks. Previous studies have demonstrated that the replacement of carbohydrates with protein is associated with better metabolic health [36]. Our findings suggest it may result in a larger effect if this replacement occurs at lunch rather than breakfast. Meanwhile, we found that moderate fat intake (26.28%–35.87% of energy) at lunch was associated with a lower increase in CMRS. This indicates that children need to consume sufficient essential fats, which are important for children’s metabolic health, growth, and development [37]. Lunch high in fish, chicken, eggs, unprocessed meat, beans, and nuts; and low in refined grains, fried foods, and sugar-sweetened beverages to provide an optimal composition of macronutrients for both growth and metabolic health in children. 

We also found that a dinner low in carbohydrate and high in protein was associated with a lower increase in CMRS, highlighting the importance of high protein intake on metabolic health. Since carbohydrates are less functional in the formation and regulation of the body’s tissues than protein and fat, diets that are relatively low in carbohydrates and high in protein have been recommended for the prevention of CMR [36]. However, dietary guidelines recommended that carbohydrates provide 45%–65% of total energy intake [38], given that excess protein and fat intake may result in harmful effects for health. Our findings suggest that the consumption of more carbohydrates at breakfast and fewer carbohydrates at lunch and dinner may be a better option to meet the minimum requirement, and prevent CMR. The macronutrient composition of snacks was not significantly associated with the change in CMRS in our study, which might be partly due to the low contribution of snacks to the total energy (10%). Our study, with a large sample size, demonstrates that a low-carbohydrate, high-protein and moderate-fat lunch; and a low-carbohydrate and high-protein dinner were associated with better metabolic health.

The strengths of the present study included the large sample size and the measurement of multiple CMR factors at both baseline and follow-up. To our knowledge, this is the first longitudinal study to comprehensively examine the association between meal patterns and change in CMRS in children. This study also had several limitations. The relative short-term follow-up (1 year) of our study may not be a long enough duration to judge the association between meal patterns and changes in CMR factors in children. Furthermore, because of the observational nature of the design of the present study, causal relations could not be established based on our findings. Thirdly, 24 h food records failed to account for seasonal variation of dietary intake, especially in fruits and vegetables; however, the dietary intakes were comparable between individuals given that all data were collected in May of the same year.

## 5. Conclusions

Meal patterns driven by energy are not significantly associated with changes in CMR factors, however, a low carbohydrate-high protein-moderate fat lunch and a low carbohydrate-high protein dinner are associated with a favorable change in CMRS in children.

## Figures and Tables

**Figure 1 nutrients-12-00799-f001:**
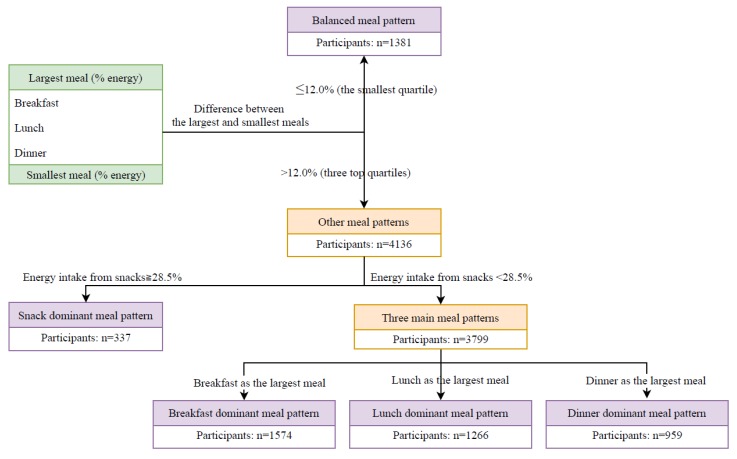
Diagram for the definition of meal patterns.

**Figure 2 nutrients-12-00799-f002:**
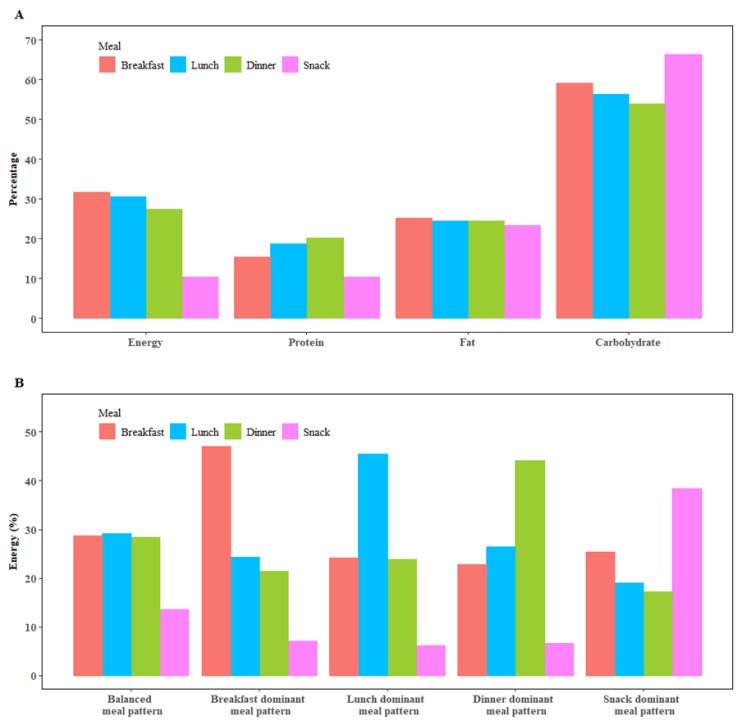
Energy and macronutrients intake from different meals. Panel A shows energy and macronutrient intake from breakfast, lunch, dinner, and snacks where macronutrient intake represents the percentage of total energy intake accounted for by the corresponding macronutrient. Panel B shows energy intake from breakfast, lunch, dinner, and snacks by meal patterns.

**Table 1 nutrients-12-00799-t001:** Baseline characteristics by meal patterns.

	Meal Pattern					*p*-Value *
	Balanced	Breakfast Dominant	Lunch Dominant	Dinner Dominant	Snack Dominant	
Age (years)	9.54 ± 1.18 ^†^	9.54 ± 1.19	9.56 ± 1.18	9.59 ± 1.19	9.38 ± 1.17	0.66
BMI (kg/m^2^)	16.98 ± 3.09	17.15 ± 3.12	17.44 ± 3.24	17.34 ± 3.40	16.90 ± 2.94	0.0413
WC (cm)	57.88 ± 8.39	58.43 ± 8.88	59.23 ± 9.11	58.87 ± 9.14	56.92 ± 8.29	0.21
PBF (%)	23.65 ± 4.90	24.00 ± 4.70	24.24 ± 4.94	23.82 ± 4.92	23.87 ± 4.49	0.25
SBP (mm Hg)	100.04 ± 10.80	100.43 ± 10.79	100.89 ± 10.49	101.25 ± 11.59	101.26 ± 10.80	0.0026
DBP (mm Hg)	64.14 ± 8.95	64.01 ± 8.88	64.10 ± 8.97	64.56 ± 9.91	64.51 ± 8.40	0.22
TC (mmol/L)	4.11 ± 0.78	4.00 ± 0.77	4.12 ± 0.78	4.18 ± 0.83	3.92 ± 0.80	0.62
HDL-C (mmol/L)	1.50 ± 0.31	1.45 ± 0.30	1.46 ± 0.30	1.48 ± 0.31	1.45 ± 0.30	0.0917
LDL-C (mmol/L)	2.13 ± 0.60	2.04 ± 0.65	2.18 ± 0.62	2.21 ± 0.66	2.06 ± 0.64	0.0100
TG (mmol/L)	0.80 ± 0.45	0.80 ± 0.42	0.84 ± 0.43	0.84 ± 0.51	0.85 ± 0.46	0.0045
Fasting glucose (mmol/L)	4.55 ± 0.54	4.50 ± 0.58	4.52 ± 0.55	4.56 ± 0.54	4.36 ± 0.61	0.0196
Log insulin	1.64 ± 0.62	1.64 ± 0.58	1.66 ± 0.64	1.74 ± 0.60	1.54 ± 0.56	0.18
Log HOMA-IR	−2.86 ± 0.66	−2.87 ± 0.62	−2.84 ± 0.68	−2.75 ± 0.63	−2.99 ± 0.62	0.32
CMRS	−0.33 ± 2.37	−0.26 ± 2.37	−0.04 ± 2.38	−0.22 ± 2.45	−0.34 ± 2.48	0.22
Physical activity (MET/week)	597.96 ± 537.13	665.70 ± 577.08	624.26 ± 577.29	626.43 ± 605.05	613.57 ± 389.78	0.76
Energy (kcal/day)	1387.47 ± 584.22	1111.51 ± 535.18	1260.94 ± 570.12	1360.68 ± 598.81	1361.39 ± 644.53	0.11
Protein intake (g/100 Kcal/day)	4.31 ± 1.04	4.30 ± 1.06	4.39 ± 1.17	4.54 ± 1.22	3.90 ± 1.04	0.62
Fat intake (g/100 Kcal/day)	2.93 ± 1.11	2.93 ± 1.07	2.86 ± 1.19	2.99 ± 1.25	3.36 ± 1.08	0.0002
Carbohydrate intake (g/100 Kcal/day)	14.25 ± 2.80	14.28 ± 2.76	14.30 ± 3.13	13.88 ± 3.16	13.70 ± 2.74	0.0003
Fibre intake (g/100 Kcal/day)	0.52 ± 0.30	0.51 ± 0.30	0.55 ± 0.37	0.54 ± 0.38	0.48 ± 0.24	0.70
Vitamin C intake (mg/100 Kcal/day)	3.21 ± 2.51	3.16 ± 2.55	3.24 ± 2.63	3.43 ± 2.83	3.36 ± 2.51	0.0317
Vitamin E intake (mg/100 Kcal/day)	0.26 ± 0.17	0.28 ± 0.19	0.25 ± 0.16	0.26 ± 0.17	0.36 ± 0.35	0.0034
Carotene intake (ug/100 Kcal/day)	76.54 ± 74.69	73.63 ± 81.28	75.26 ± 86.98	81.69 ± 95.16	75.02 ± 71.97	0.31
Magnesium intake (mg/100 Kcal/day)	14.78 ± 3.43	14.96 ± 3.54	15.15 ± 3.92	15.09 ± 4.16	14.47 ± 3.68	0.42
Potassium intake (mg/100 Kcal/day)	99.25 ± 29.71	100.61 ± 30.67	98.93 ± 32.49	101.62 ± 35.67	103.58 ± 31.28	0.0453
Phosphorus intake (mg/100 Kcal/day)	60.66 ± 12.77	62.48 ± 13.14	61.06 ± 13.46	62.58 ± 14.55	57.38 ± 13.74	0.32
Calcium intake (mg/100 Kcal/day)	30.29 ± 13.97	32.76 ± 16.83	27.43 ± 13.46	27.95 ± 13.82	34.97 ± 18.91	0.0203
Iron intake (mg/100 Kcal/day)	1.22 ± 0.68	1.12 ± 0.55	1.29 ± 0.89	1.21 ± 0.63	1.05 ± 0.50	0.50
Sex						0.83
Boys	668 (48.4) ^‡^	791 (50.3)	628 (49.6)	481 (50.2)	152 (45.1)	
Girls	713 (51.6)	783 (49.7)	638 (50.4)	478 (49.8)	185 (54.9)	
Grade						0.60
Two	390 (28.2)	442 (28.1)	363 (28.7)	260 (27.1)	111 (32.9)	
Three	373 (27.0)	475 (30.2)	301 (23.8)	253 (26.4)	101 (30.0)	
Four	365 (26.4)	422 (26.8)	377 (29.8)	260 (27.1)	90 (26.7)	
Five	253 (18.3)	235 (14.9)	225 (17.8)	186 (19.4)	35 (10.4)	
Puberty						0.07
Yes	1275 (92.3)	1470 (93.4)	1165 (92.0)	862 (89.9)	312 (92.6)	
No	106 (7.7)	104 (6.6)	101 (8.0)	97 (10.1)	25 (7.4)	

BMI, body mass index; CMRS, cardiometabolic risk score; DBP, diastolic blood pressure; HDL-C, high-density lipoprotein cholesterol; HOMA-IR, homeostatic model assessment of insulin resistance; LDL-C, low-density lipoprotein cholesterol; MAP, mean arterial pressure; PBF, percent body fat; SBP, systolic blood pressure; TC, total cholesterol; TG, triglyceride; WC, waist circumference. * ANOVA was used to test the difference of continuous variables across meal patterns and chi-square test for categorical variables. ^†^ All such data were mean ± standard deviation. ^‡^ All such data were frequency (%).

**Table 2 nutrients-12-00799-t002:** Meal patterns and changes in cardiometabolic risk factors in children.

	Meal Pattern					*p*-Value *
	Balanced	Breakfast Dominant	Lunch Dominant	Dinner Dominant	Snack Dominant	
Change in BMI						
Participants	1367	1553	1249	945	334	
β (95% CI) ^†^		0.032 (−0.0114, 0.076)	0.039 (−0.006, 0.084)	0.003 (−0.045, 0.052)	0.010 (−0.061, 0.081)	0.37
Change in WC						
Participants	1360	1549	1245	945	334	
β (95% CI)		−0.012 (−0.046, 0.022)	0.032 (−0.003, 0.067)	−0.006 (−0.044, 0.032)	0.012 (−0.043, 0.068)	0.13
Change in PBF						
Participants	1337	1510	1209	915	326	
β (95% CI)		0.042 (−0.011, 0.095)	0.012 (−0.043, 0.067)	−0.035 (−0.094, 0.024)	−0.069 (−0.156, 0.018)	0.0320
Change in SBP						
Participants	1361	1551	1246	942	333	
β (95% CI)		0.054 (−0.015, 0.123)	0.046 (−0.025, 0.117)	0.057 (−0.019, 0.133)	−0.060 (−0.173, 0.052)	0.17
Change in DBP						
Participants	1363	1552	1248	943	334	
β (95% CI)		0.023 (−0.047, 0.094)	0.007 (−0.066, 0.079)	−0.013 (−0.092, 0.065)	−0.057 (−0.172, 0.058)	0.68
Change in MAP						
Participants	1361	1550	1246	943	334	
β (95% CI)		0.038 (−0.032, 0.108)	0.026 (−0.046, 0.098)	0.016 (−0.061, 0.094)	−0.053 (−0.167, 0.061)	0.56
Change in TC						
Participants	1283	1460	1175	892	316	
β (95% CI)		−0.062 (−0.115, −0.008)	0.001 (−0.054, 0.056)	−0.025 (−0.084, 0.035)	0.043 (−0.044, 0.130)	0.0513
Change in HDL-C						
Participants	1284	1459	1175	891	314	
β (95% CI)		0.027 (−0.049, 0.103)	−0.066 (−0.144, 0.012)	−0.027 (−0.1107, 0.057)	0.270 (0.146, 0.393)	<0.0001
Change in LDL-C						
Participants	1284	1461	1176	891	316	
β (95% CI)		−0.051 (−0.108, 0.006)	−0.063 (−0.122, −0.004)	−0.037 (−0.100, 0.026)	0.004 (−0.089, 0.096)	0.21
Change in TG						
Participants	1282	1461	1176	894	317	
β (95% CI)		−0.029 (−0.098, 0.039)	0.001 (−0.070, 0.071)	0.017 (−0.059, 0.093)	−0.193 (−0.304, −0.082)	0.0075
Change in fasting glucose						
Participants	1284	1460	1176	892	317	
β (95% CI)		0.030 (−0.028, 0.088)	−0.047 (−0.107, 0.013)	0.015 (−0.050, 0.079)	0.040 (−0.054, 0.135)	0.11
Change in insulin						
Participants	1132	1278	1035	795	273	
β (95% CI)		−0.048 (−0.156, 0.059)	−0.055 (−0.166, 0.055)	0.119 (−0.0001, 0.237)	−0.125 (−0.300, 0.051)	0.0181
Change in HOMA-IR						
Participants	1132	1277	1034	795	273	
β (95% CI)		−0.036 (−0.142, 0.069)	−0.069 (−0.177, 0.039)	0.114 (−0.002, 0.230)	−0.104 (−0.275, 0.068)	0.0199
Change in CMRS ^‡^						
Participants	1179	1331	1066	798	300	
β (95% CI)		0.059 (−0.107, 0.225)	0.113 (−0.058, 0.284)	0.079 (−0.105, 0.264)	−0.324 (−0.590, −0.058)	0.031

BMI, body mass index; CI, confidence interval; CMRS, cardiometabolic risk score; DBP, diastolic blood pressure; HOMA-IR, homeostatic model assessment of insulin resistance; HDL-C, high-density lipoprotein cholesterol; LDL-C, low-density lipoprotein cholesterol; MAP, mean arterial pressure; PBF, percent body fat; SBP, systolic blood pressure; TC, total cholesterol; TG, triglyceride; WC, waist circumference. * GLM was used to estimate multivariable-adjusted β (95% CI) of cardiometabolic risk factors between-meal patterns with the balanced meal pattern as the reference. We used the Benjamin–Hochberg procedure to control the false discovery rate at level 5% for multiple comparisons with the P-value cut-off point of significance was 0.0071. ^†^ Multivariable analysis was adjusted for classes in school as clustering effects and characteristics of individuals including age, sex, corresponding CMR factor at baseline, puberty, grade, intervention, BMI, physical activity, energy intake, protein intake, carbohydrate intake, fat intake, fiber intake, vegetable intake, fruit intake, pork intake, legumes intake, nuts intake, birthweight, household income, mother’s education, father’s education, mother’s BMI, and father’s BMI as fixed effects. ^‡^ CMRS was calculated by summing Z scores of WC, the average of SBP and DBP, fasting glucose, HDL-C (multiplying by –1), and TG.

**Table 3 nutrients-12-00799-t003:** Energy intake from meals and snacks and change in cardiometabolic risk score *.

	Consumption Level					*p-Trend* ^†^
	Quintile 1	Quintile 2	Quintile 3	Quintile 4	Quintile 5	
Energy from breakfast						
Range (%)	<21.06	21.06–27.39	27.40–33.40	33.41–41.67	>41.67	
Participants	940	930	933	925	946	
β (95% CI), Model 1 ^‡^		0.019 (−0.179, 0.216)	0.036 (−0.161, 0.234)	0.193 (−0.006, 0.391)	0.042 (−0.158, 0.242)	0.32
β (95% CI), Model 2 ^§^		0.041 (−0.147, 0.228)	0.026 (−0.162, 0.213)	0.202 (0.013, 0.391)	0.049 (−0.144, 0.241)	0.24
β (95% CI), Model 3 ^¶^		0.044 (−0.143, 0.232)	0.023 (−0.164, 0.210)	0.189 (0.0002, 0.377)	0.043 (−0.149, 0.236)	0.31
Energy from lunch						
Range (%)	<20.81	20.81–27.07	27.08–32.59	32.60–39.09	>39.09	
Participants	954	942	920	932	926	
β (95% CI), Model 1		0.037 (−0.161, 0.235)	0.080 (−0.122, 0.283)	0.135 (−0.068, 0.338)	0.248 (0.044, 0.452)	0.14
β (95% CI), Model 2		0.016 (−0.172, 0.203)	0.100 (−0.091, 0.292)	0.138 (−0.054, 0.330)	0.194 (0.0002, 0.387)	0.24
β (95% CI), Model 3		0.035 (−0.152, 0.223)	0.091 (−0.1001, 0.283)	0.143 (−0.049, 0.335)	0.173 (−0.021, 0.367)	0.38
Energy from dinner						
Range (%)	<18.08	18.09–24.31	24.32–29.88	29.89–36.64	>36.64	
Participants	957	943	913	934	927	
β (95% CI), Model 1		0.185 (−0.012, 0.381)	0.167 (−0.032, 0.367)	−0.027 (−0.225, 0.172)	0.015 (−0.184, 0.213)	0.10
β (95% CI), Model 2		0.128 (−0.058, 0.314)	0.126 (−0.063, 0.315)	−0.049 (−0.237, 0.139)	0.010 (−0.179, 0.199)	0.23
β (95% CI), Model 3		0.136 (−0.051, 0.322)	0.145 (−0.045, 0.335)	−0.029 (−0.218, 0.159)	0.016 (−0.173, 0.205)	0.22
Energy from snacks						
Range (%)	0	0–2.47	2.48–8.32	8.33–19.03	>19.03	
Participants	1399	457	923	944	951	
β (95% CI), Model 1		−0.069 (−0.303, 0.164)	−0.037 (−0.219, 0.145)	−0.158 (−0.340, 0.024)	−0.101 (−0.284, 0.082)	0.51
β (95% CI), Model 2		−0.003 (−0.224, 0.2184)	−0.0140 (−0.1872, 0.1593)	−0.1305 (−0.3058, 0.0448)	−0.0827 (−0.2700, 0.1046)	0.61
β (95% CI), Model 3		0.018 (−0.204, 0.239)	−0.010 (−0.183, 0.164)	−0.102 (−0.277, 0.074)	−0.064 (−0.251, 0.124)	0.77

CI, confidence interval. * Cardiometabolic risk score was calculated by summing Z scores of waist circumference, the average of systolic and diastolic blood pressure, fasting glucose, high-density lipoprotein cholesterol (multiplying by –1) and triglyceride. ^†^ GLM was used to estimate multivariable-adjusted β (95% CI) of cardiometabolic risk score between quintiles of energy intake from different meals with the quintile 1 as the reference. ^‡^ Model 1 was adjusted for classes in school as clustering effects and characteristics of individuals, including age, sex, and corresponding CMR factor at baseline as fixed effects. ^§^ Model 2 was adjusted for Model 1 plus puberty, grade, intervention, puberty, BMI, physical activity, and total energy intake. ^¶^ Model 3 was adjusted for Model 2 plus birthweight, household income, mother’s education, father’s education, mother’s BMI, and father’s BMI as fixed effects.

**Table 4 nutrients-12-00799-t004:** Macronutrient intake from meals and snacks and change in cardiometabolic risk score*.

	Consumption Level					*p-Trend* ^†^
	Quintile 1	Quintile 2	Quintile 3	Quintile 4	Quintile 5	
Carbohydrate at breakfast						
Range (% energy)	<47.54	47.54–56.39	56.40–63.19	63.20–72.05	>72.05	
Participants	948	924	920	946	936	
β (95% CI) ^‡^		−0.032 (−0.228, 0.163)	−0.128 (−0.337, 0.080)	−0.194 (−0.419, 0.030)	−0.054 (−0.310, 0.202)	0.31
Carbohydrate at lunch						
Range (% energy)	<41.24	41.24–52.63	52.63–62.15	62.16–72.16	>72.16	
Participants	923	920	922	938	971	
β (95% CI)		0.217 (0.018, 0.416)	0.315 (0.098, 0.531)	0.603 (0.368, 0.837)	0.777 (0.509, 1.046)	<0.0001
Carbohydrate at dinner						
Range (% energy)	<36.61	36.61–49.79	49.80–60.70	60.71–72.70	>72.70	
Participants	927	927	939	944	937	
β (95% CI)		0.345 (0.145, 0.545)	0.601 (0.383, 0.818)	0.662 (0.428, 0.907)	0.663 (0.387, 0.938)	<0.0001
Carbohydrate from snacks						
Range (% energy)		0–41.12	41.13–63.92	63.93–82.98	>82.98	
Participants	1393	469	922	961	929	
β (95% CI)		0.048 (−0.174, 0.269)	−0.108 (−0.287, 0.072)	−0.117 (−0.294, 0.060)	0.077 (−0.097, 0.251)	0.18
Protein at breakfast						
Range (% energy)	<11.83	11.83–13.77	13.78–15.76	15.77–18.61	>18.61	
Participants	954	919	947	929	925	
β (95% CI)		0.010 (−0.177, 0.196)	0.026 (−0.169, 0.220)	0.048 (−0.161, 0.257)	−0.165 (−0.417, 0.086)	0.30
Protein at lunch						
Range (% energy)	<12.78	12.78–15.84	15.85–19.14	19.15–23.92	>23.92	
Participants	959	960	931	921	903	
β (95% CI)		0.025 (−0.162, 0.213)	0.143 (−0.054, 0.339)	−0.263 (−0.474, −0.052)	−0.4632 (−0.710, −0.217)	<0.0001
Protein at dinner						
Range (% energy)	<13.01	13.01–16.86	16.87–20.75	20.76–26.47	>26.47	
Participants	933	934	933	946	928	
β (95% CI)		0.080 (−0.114, 0.273)	−0.048 (−0.248, 0.153)	0.102 (−0.111, 0.314)	−0.360 (−0.602, −0.117)	<0.0001
Protein from snacks						
Range (% energy)	0	0–3.95	3.96–8.14	8.15–12.40	>12.40	
Participants	1412	457	921	940	944	
β (95% CI)		−0.041 (−0.263, 0.180)	−0.043 (−0.219, 0.134)	−0.144 (−0.320, 0.033)	0.064 (−0.113, 0.241)	0.27
Fat at breakfast						
Range (% energy)	<14.23	14.23–21.35	21.35–27.59	27.60–35.20	>35.20	
Participants	935	940	925	922	952	
β (95% CI)		−0.106 (−0.296, 0.085)	−0.019 (−0.221, 0.183)	−0.030 (−0.243, 0.183)	0.086 (−0.158, 0.329)	0.39
Fat at lunch						
Range (% energy)	<11.13	11.13–18.98	18.99–26.27	26.28–35.87	>35.87	
Participants	976	925	923	916	934	
β (95% CI)		−0.260 (−0.453, −0.067)	−0.296 (−0.504, −0.089)	−0.507 (−0.726, −0.289)	−0.441 (−0.685, −0.197)	0.0003
Fat at dinner						
Range (% energy)	<9.75	9.75–17.95	17.96–26.72	26.73–37.30	>37.30	
Participants	947	940	930	922	935	
β (95% CI)		0.169 (−0.027, 0.365)	0.110 (−0.101, 0.321)	−0.017 (−0.244, 0.210)	−0.146 (−0.400, 0.109)	0.10
Fat from snacks						
Range (% energy)		0–3.26	3.27–16.78	16.79–33.34	>33.34	
Participants	1426	443	926	939	940	
β (95% CI)		0.040 (−0.184, 0.264)	0.006 (−0.168, 0.180)	−0.029 (−0.206, 0.149)	−0.068 (−0.244, 0.107)	0.89

CI, confidence interval; CMRS, cardiometabolic risk score * CMRS was calculated by summing Z scores of waist circumference, the average of systolic and diastolic blood pressure, fasting glucose, high-density lipoprotein cholesterol (multiplying by –1) and triglyceride. ^†^ GLM was used to estimate multivariable-adjusted β (95% CI) of cardiometabolic risk score between quintiles of macronutrient intake from different meals with the quintile 1 as the reference. We used the Benjamin–Hochberg procedure to control the false discovery rate at level 5% for multiple comparisons with the P-value cut-off point of significance was 0.0208. ^‡^ Multivariable analysis was adjusted for classes in school as clustering effects and characteristics of individuals including age, sex, and CMRS at baseline, puberty, grade, intervention, puberty, BMI, physical activity, and total energy intake, intake of grains, fried foods, vegetable, fruit, nuts, pork, red meat rather than pork, poultry, eggs, milk, and sugar-sweetened beverage, birthweight, household income, mother’s education, father’s education, mother’s BMI, and father’s BMI as fixed effects.

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
