# Peer review of "Meal Patterns and Changes in Cardiometabolic Risk Factors in Children: A Longitudinal Analysis"

_nutrients, 2020, doi:10.3390/nu12030799_

Round 1

Reviewer 1 Report

The paper entitled “Meal patterns and changes in cardiometabolic risk factors in children: a longitudinal analysis” is a paper about a crucial public health issue concerning child health. This is a well-written article based on a large general population-based study including 5517 children, although there are some concerns about several issues that the authors should carefully tackle before considering the manuscript for publication.

1) As reported, the aim of the present study was to examine the association of different meal patterns driven by energy intake; however, there is no clear explanation about how these meal patterns were derived. Please, provide more information about this issue.

2) Another important issue of the study was to analyze the changes in cardiometabolic risk (CMR) factors. Based on that the authors reported, they gathered information at baseline (May 2009) and at follow-up, one year after (May 2010). To examine changes between these time-points, they evaluated the difference in changes in CMR factors using linear regression models and displayed results as Mean ± SE. However, it should be noted that main estimate of linear regression analysis is the beta, not the mean. Moreover, the symbol “±” used to describe the results should be avoided. It is preferable to present the measures of variability in brackets or as confidence intervals (please, the authors can see the following references):

  1. Altman DG, Gore SM, Gadner MJ, Pocock SJ. Statistical guidelines for contributors to medical journals. Br Med J 1983;286: 1.489-1.493
  2. Bailar JC, Mosteller F. Guidelines for statistical reporting in articles for medical journals: amplifications and explanations. Ann Intern Med 1988;108: 266-273

Minor issues:

1) All the acronyms used in tables should be defined clearly.

Reviewer 2 Report

Well done- this is a very comprehensive paper with good methodology. I would suggests that  the authors consider moving some of the tables to supplementary data, especially those that don't add significantly to the conversation as there is a lot of tables and data to go though. One also has to be cautious about over interpreting one positive p due to multiplicity testing. This applies to the energy at lunch- see line 162. That line contradicts the first assertion that energy driven meal patterns were not associated with CMR factors. Overall

Minor comments- i have bolded change required 

line 18- whether energy and macronutrient intake from different meals was- 

line 37- CMR factors are highly likely to persist into adulthood and are associated 

line 43-snacks per day is a norm and may be a

line 43-44- makes no sense

line 49- remove meanwhile

line 79-80-We defined a balanced meal pattern as the difference between the largest and smallest meals was 12.0% (smallest quartile) or less of total energy intake.We defined a balanced meal pattern as ≤12.0% (smallest quartile) difference between the largest and smallest meals or of total energy intake.

Round 2

Reviewer 1 Report

The authors have addressed all the issues required by the reviewers properly. To facilitate the reading of tables, it is suggested to delete reference category including a note at the end of the table. Moreover, it is recommended to shorten the decimals as much as possible. 

Author Response

The authors have addressed all the issues required by the reviewers properly. To facilitate the reading of tables, it is suggested to delete reference category including a note at the end of the table. Moreover, it is recommended to shorten the decimals as much as possible.

Response:

As suggested by the Reviewer, we have deleted the reference category and included a note for Tables 2-4.

We have also shortened the decimals accordingly.